# DigGAN: Discriminator gradIent Gap Regularization for GAN Training with Limited Data

**Tiantian Fang, Ruoyu Sun, Alex Schwing**
University of Illinois Urbana-Champaign
{tf6, ruoyus, aschwing}@illinois.edu

## Abstract

Generative adversarial nets (GANs) have been remarkably successful at learning to sample from distributions specified by a given dataset, particularly if the given dataset is reasonably large compared to its dimensionality. However, given limited data, classical GANs have struggled, and strategies like output-regularization, data-augmentation, use of pre-trained models and pruning have been shown to lead to improvements. Notably, the applicability of these strategies is 1) often constrained to particular settings, e.g., availability of a pretrained GAN; or 2) increases training time, e.g., when using pruning. In contrast, we propose a Discriminator gradIent Gap regularized GAN (DigGAN) formulation which can be added to any existing GAN. DigGAN augments existing GANs by encouraging to narrow the gap between the norm of the gradient of a discriminator's prediction w.r.t. real images and w.r.t. the generated samples. We observe this formulation to avoid bad attractors within the GAN loss landscape, and we find DigGAN to significantly improve the results of GAN training when limited data is available. Code is available at `https://github.com/AilsaF/DigGAN`.

## 1 Introduction

Generative Adversarial Nets (GANs) [13] have been remarkably successful at learning to sample from distributions specified by a given dataset. In practice, this success has garnered a lot of interest in GANs for a wide range of applications, from data augmentation [23, 61] and domain adaptation [8, 48] to image-to-image translation [63, 18, 24] and photo editing [4, 64].

This success of GANs strongly relies on the availability of a large dataset. Unsurprisingly, in real-life circumstances, particularly when the dimensionality of the samples in the dataset is high, the available samples to train a GAN can be insufficient. Insufficient data may significantly reduce the performance of standard GANs. For instance, when we train a GAN on CIFAR-100 using just $10\%$ of the data, BigGAN performance deteriorates from 13.54 FID score to 73.01 FID score, and the GAN generates images of a single pattern (Fig. 1). To address this deteriorating performance of GANs trained with limited data, various strategies have been proposed recently, including use of a pretrained model [60, 42, 28], pruning [7], and data augmentation [23, 61, 59, 62, 19]. However, despite improving results, each of these strategies also impose restrictions. The use of pretrained models works best if data domains remain similar. Pruning requires many rounds of training to increase the sparsity of the neural architecture, which raises the training cost. Data augmentation can enhance the results, but the benefit is limited with insufficient data (Tab. 3). Regularization is a cheap and potentially effective approach, and recent work by Tseng et al. [51] adopted this approach, controlling the distance between the discriminator's prediction on the real image and the generated image. However, with limited data, this regularization doesn't show significant improvements (Tab. 3).

In this paper, we study a new regularization to enhance the training of GANs with limited data. Instead of constraining the discriminator's output as done in prior work [51], we propose the Discriminator

36th Conference on Neural Information Processing Systems (NeurIPS 2022).

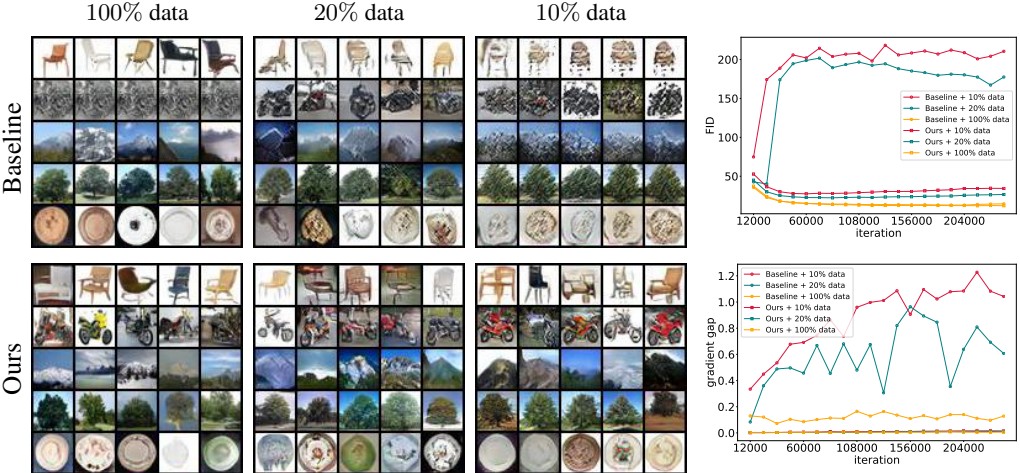

Figure 1: CIFAR-100 generated samples by BigGAN (baseline) and BigGAN+DigGAN (ours), both trained with 100%, 20% and 10% data at the iteration with the best FID (left). We observe: 1) when training with 10% data, quantitative scores of the baseline deteriorate considerably (top right) and the gradient norm gap increases (bottom right). The gradient norm gap size and FID are inversely proportional to the amount of training data. 2) The proposed DigGAN regularization stabilizes both quantitative scores and the gradient norm gap.

gradIent Gap GAN (DigGAN) regularization. Though multiple gradient-based regularizers have been proposed for GAN training [15, 37, 26], none of these target training of GANs with limited data. Instead, they focus on training stability of GANs with standard data.

In contrast to prior formulations, the DigGAN regularizer encourages to narrow the gap between 1) the norm of the gradient of a discriminator's prediction w.r.t. real images, and 2) the norm of the gradient of a discriminator's prediction w.r.t. generated data. For readability, we call this gap "DIG" (Discriminator gradIent Gap). Our regularizer is motivated by empirical studies, during which we found that: 1) When training GANs with limited data, the DIG is large. 2) By reducing the DIG, the effects of bad attractors in the training dynamics of GANs can be reduced. We found this to improve training by preventing GANs from being pushed to bad attractors.

We conduct comprehensive experiments to show the effectiveness and consistency of the proposed regularizer. First, we use a synthetic example to demonstrate the stabilizing effect of the proposed regularizer on the training, avoiding bad attractors that the original GAN is attached to. Second, using recent architectures that achieve state-of-the-art results, including SNGAN, BigGAN, and StyleGAN2, we show that the regularizer improves results when only limited data is available.

## 2  Related Work

In the following we briefly discuss GAN variants, regularization for GANs, and prior work to address training of GANs with limited data.

**Generative Adversarial Networks.** Many GAN variants have been proposed to stabilize the training and improve the perceptual quality of the generated results. Among them, many variants study different loss objectives [14, 35, 41, 31, 3, 43, 1, 15, 54, 39, 40, 20, 50, 11, 10, 32]. The design of new architectures is another popular direction to obtain different GAN variants [44, 12, 63, 5, 22, 21]. Moreover, a variety of normalization techniques have also been proposed [46, 58, 38, 17, 2]. Finally, techniques have also been devised to produce more diverse samples [33, 55] and to improve convergence [36, 57, 34, 47].

**Regularization for GANs.** Among all the aforementioned GANs, regularization techniques are widely used to stabilize the training. Gulrajani et al. [15] regularize the discriminator to be 1-Lipschitz by penalizing the norm of the gradient of the discriminator with respect to its input data points sampled by interpolating between real data and generated data. Kodali et al. [26] propose to penalize only the Gaussian manifold around the real data instead. Roth et al. [45] encourage the gradient norm of the

discriminator on real data and generated data to be zero. Mescheder et al. [37] provide more detailed insights on the gradient penalty. Besides the gradient norm, constraining the discriminator is another common mechanism [59, 62] and the weight penalty is also a common regularization method for GANs [4, 5, 29]. Different from these methods, we focus on improving results in the case of limited training data.

**Training of GANs with limited data.** Training GANs with limited data has attracted a lot of attention recently. Data scarcity causes classical GAN training to become more challenging [23, 53, 49]. A few methods have been proposed to improve the performance of GANs trained with limited data. One popular idea is data augmentation [62, 59, 23, 61, 19, 6]. Jiang et al. [19] use generated data as an "augmentation" for the real data, while others do the augmentation on real instances. Chen et al. [7] leverage pruned neural networks to improve the performance. A few works [60, 42, 28] used pre-trained GANs, which were learned on a similar data domain using sufficient data before being transferred to the target domain. Yang et al. [56] require the discriminator to do instance-level classification and distinguish every individual real and fake image instance as an independent category. Most closely related to our method is work by Tseng et al. [51], who regularize the distance between the discriminator evaluated on the real data and the discriminator evaluated on the generated data to be small. Our approach differs from these methods in that we propose a new regularizer which encourages the Discriminator gradIent Gap of a GAN to be small. This regularizer is orthogonal to other solutions (except [51]) and may be used concurrently.

## 3 Method

### 3.1 Review of Generative Adversarial Networks

Generative adversarial networks (GANs) consist of a generator $G$ and a discriminator $D$, which are pitted against each other. The generator $G(z; \theta)$, parameterized by $\theta$, learns to map a sample $z \sim \mathcal{Z}$ drawn from a simple, possibly low-dimensional distribution $p(z)$ (e.g., Gaussian) over domain $\mathcal{Z}$, to the complex, possibly high-dimensional data distribution domain $\mathcal{X}$. The discriminator $D(x)$ is trained to distinguish between real data $x_R \sim \mathcal{X}$ and synthetically generated data $x_F = G(z; \theta)$ obtained from the generator. More formally, the game between the generator $G$ and the discriminator $D$ is described by the following two loss functions, which are often also referred to as the non-saturating GAN losses:

$$
\begin{aligned}
\mathcal{L}_G &= \mathbb{E}_{z \sim p(z)}[\ell_G(-D(G(z; \theta)))], \\
\mathcal{L}_D &= \mathbb{E}_{x \sim p_{\text{data}}(x)}[\ell_D(-D(x))] + \mathbb{E}_{z \sim p(z)}[\ell_D(D(G(z; \theta)))].
\end{aligned}
\tag{1}
$$

Multiple loss functions have been used for $\ell_G$ and $\ell_D$. For instance, Jenson-Shannon-GAN [14] uses $\ell_G(t) = \ell_D(t) = \log(1 + \exp(t))$, and hingeGAN [9] uses $\ell_G(t) = t$ and $\ell_D(t) = \max(0, 1 + t)$.

### 3.2 Discriminator GradIent Gap (DigGAN)

**Worse Performance of GANs with Limited Data.** Karras et al. [23] and Tseng et al. [51] observed that fewer training samples can cause GANs to perform much worse. In empirical studies (e.g., Fig. 1), we observe this too. Concretely, we train a BigGAN with 100%, 20% and 10% CIFAR-100 data. The FID score (lower is better) increases dramatically when 10% or 20% data are used (top right part of Fig. 1). Subsequently the generator produces images of worse quality (see the images shown in the first row of Fig. 1).

**Observation: large gap of gradient norms.** In comprehensive experiments across multiple datasets we observe that the gap between 1) the norm of the gradient of a discriminator's prediction w.r.t. real images, and 2) the norm of the gradient of a discriminator's prediction w.r.t. generated data, i.e., the squared distance

$$
R(D, x_R, x_F) = \left( \left\| \frac{\partial D}{\partial x_R} \right\|_2 - \left\| \frac{\partial D}{\partial x_F} \right\|_2 \right)^2,
\tag{2}
$$

increases when the GAN is trained with fewer data. Here, $x_R$ and $x_F = G(z; \theta)$ denote the training data and the generated data respectively. For readability, we refer to this gap via "DIG" (Discriminator gradIent Gap). This phenomenon can be observed in Fig. 1 (see the bottom right part) and Fig. 6 (see the right part).

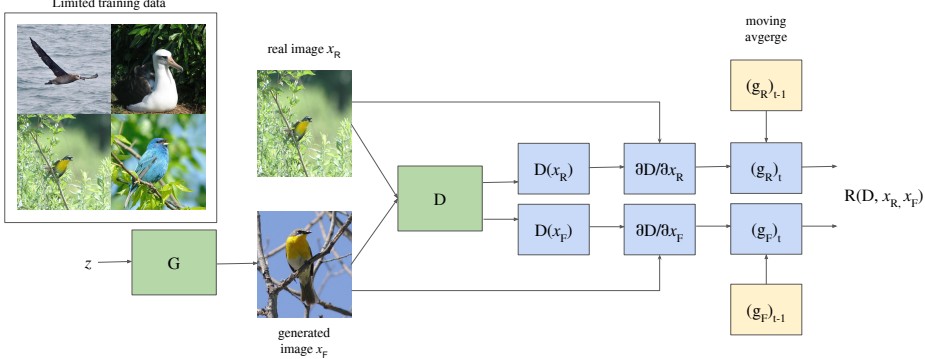

Figure 2: The DigGAN computation graph for the regularizer $R(D, x_R, x_F)$ given in Eq. (3).

Motivated by the observation, we propose a possible explanation for the training failure of GANs when limited data is available:

$$\text{Few data} \overset{(a)}{\to} \text{large "Discriminator gradIent Gap"} \overset{(b)}{\to} \text{unstable training of GANs.}$$

In this chain, $(a)$ is empirically observed in Fig. 1 and Fig. 6, and $(b)$ is due to the explanation below.

We suspect that a large gap in discriminator gradient norms (DIG) is one reason for the bad performance of GANs trained with limited data. We provide an intuition here. We can treat the gradient as the rate with which the discriminator's output changes when the input image changes, and the norm as the magnitude of this rate. A large gap between two norms can then be interpreted as a significant imbalance between 1) the discriminator's rate of change w.r.t. a real image change and 2) the discriminator's rate of change w.r.t. a generated image change. Consequently, the discriminator learns faster on either real images or generated images resulting in an imbalance.

To improve the training of GANs with limited data, it is natural to reduce the DIG. We propose to use Eq. (2) as a regularizer so as to control the DIG during training. In turn, this aids to balance the discriminator's learning speed.

More specifically, we train the generator $G$ using the objective $\mathcal{L}_G$ given in Eq. (1), but train the discriminator $D$ with the extra regularizer stated in Eq. (2), modified to account for the possible issue that data within different mini-batches can be extremely different, causing the regularizer to vary a lot. To reduce the variance, we use a moving average for the gradient norm term in the regularization, with a smoothness factor $\alpha$. Formally, when training the discriminator $D$ we use the following regularized objective at iteration $t$:

$$\tilde{\mathcal{L}}_D = \mathcal{L}_D + \lambda R(D, x_R, x_F) \quad \text{with} \quad R(D, x_R, x_F)_t = ((g_R)_t - (g_F)_t)^2, \text{where}$$

$$(g_\beta)_t = \begin{cases} \left( \left\| \frac{\partial D}{\partial x_\beta} \right\|_2 \right)_t & \text{if } t = 1, \\ (1-\alpha) \cdot (g_\beta)_{t-1} + \alpha \cdot \left( \left\| \frac{\partial D}{\partial x_\beta} \right\|_2 \right)_t & \text{if } t > 1. \end{cases} \tag{3}$$

Here, $\lambda$ is a non-negative, scalar hyper-parameter, $\alpha \in (0, 1]$ is a scalar hyper-parameter lying between 0 and 1 (inclusive), and $\beta \in \{R, F\}$ is used to refer to real or generated data. This process to compute the proposed regularizer is summarized in Fig. 2.

### 3.3 Small-Scale Case Study: Intuition for DigGAN Regularizer and Attractors

In this section, we analyze the advantage of the proposed regularization $R$ from the perspective of attractors. For readability, we refer to the local attractors of unregularized GANs as "unregularized-local-attractors." Prior work [50] argued that GAN training has sub-optimal unregularized-local-attractors, and this issue becomes less severe when there are more samples. In fact, as the number of samples goes to infinity, the empirical loss converges to a population loss which is convex [14, 50]. Consequently as the number of samples decreases, we expect the issue to become more severe. We will show that the proposed DigGAN can alleviate this issue at least on a simple example.

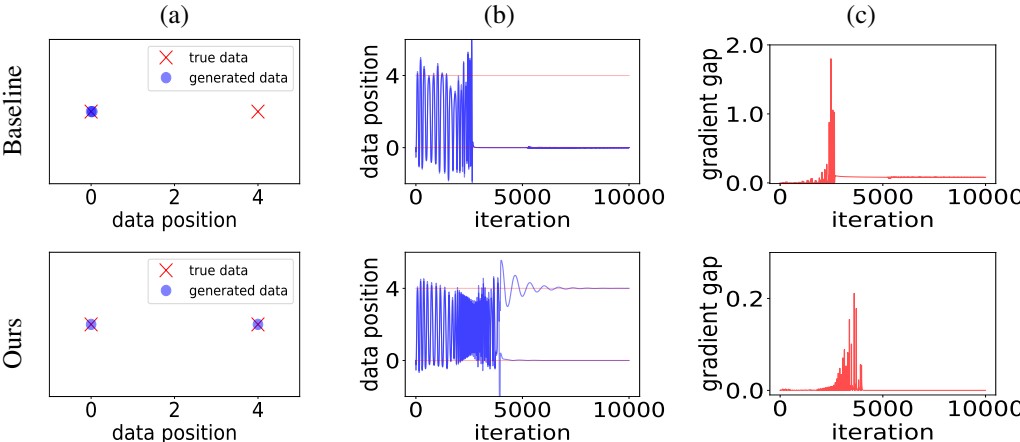

Figure 3: Comparing GAN without (top row) and with (bottom row) the proposed DigGAN regularization in a two-point experiment. (a) Data position after the training. Baseline covers one true data point, while the proposed DigGAN regularization enables a GAN to cover all true data. (b) generated data (blue curves) movement for the whole training. (c) Gradient gap $R$ (Eq. (2)).

First, we empirically show that a vanilla GAN can get stuck in local attractors (Experiment 1). Second, we show that these local attractors are stable for vanilla GAN and cannot be escaped from (Experiment 2). Third, we show that DigGAN can improve the training by avoiding unregularized-local-attractors (Experiment 3). Forth, to understand why DigGAN can avoid them, we empirically show that DigGAN can escape even if starting from unregularized-local-attractors (Experiment 4). The experiment of "escaping unregularized-local-attractors" demonstrates: the unregularized-local-attractors are no longer attractors for DigGAN. Thus DigGAN training doesn't get stuck at unregularized-local-attractors, which provides an explanation for Experiment 3. We summarize the logic for clarity:

> In DigGAN, unregularized-local-attractors can be escaped (Experiment 4). $\implies$ They are no longer local attractors of DigGAN training (by definition of local attractors). $\implies$ DigGAN training will not get trapped in unregularized-local-attractors (Experiment 3) which trap a vanilla GAN (Experiment 1 and 2).

Note that the four experiments serve different roles. The "avoiding experiment" (Experiment 3) is what naturally happens in practice, but it is not that clear how a regularizer can help avoid attractors. The "escaping experiment" (Experiment 4) is carefully designed and rarely happens in practice with random initialization (since DigGAN would not get trapped starting from a random initialization). But the experiment has two advantages: (i) results easily demonstrate the power of the regularizer; (ii) results illustrate the underlying mechanism of the "avoiding experiment".

For the four experiments, we consider the task of generating two real data points $0$ and $4$. The generator is a 4-layer fully connected network (FCN) with 5 neurons per layer and `tanh` activation. The discriminator is a 4-layer FCN with 10 neurons per layer and sigmoid activation. Both the discriminator $D$ and the generator $G$ are updated with momentum coefficient $0.9$ [25] and learning rate $0.01$. We use $\lambda = 1$ and $\alpha = 1$ in Eq. (3). We use $10,000$ iterations to complete the training and study each of the four experiments:

**Experiment 1: vanilla GAN can get stuck in unregularized-local-attractors.** We train a vanilla GAN from scratch with a random initialization $D_0$ and $G_0$. We get $D_1$ and $G_1$ at the end of training. In Fig. 3 top row, we plot (a) the final training snippet, (b) the 1-dimensional data movement, and (c) the gradient norm gap $R$ given in Eq. (2) throughout the entire training process. In plot (b), the y-axis represents the data position, while the x-axis represents the iteration. The movement of the generated data is represented via the blue curves, and all true data is represented by two straight red lines at the y-axis positions 0 and 4. We observe that the vanilla GAN (top row in Fig. 3) gets stuck generating data from a single-mode $x = 0$ (top row (a)) starting with roughly iteration 3000 (top row (b)). The baseline's gradient gap (top row (c)) increases significantly right before GAN training is trapped.

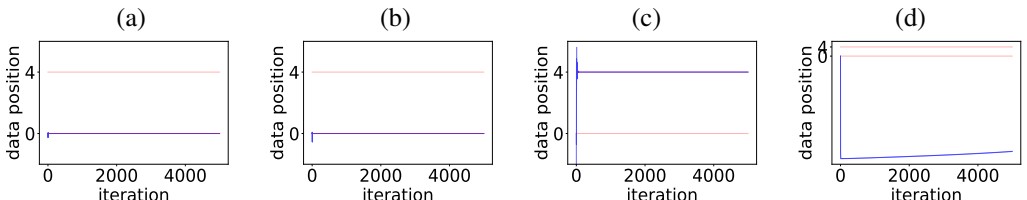

Figure 4: Discriminator perturbation experiments. Adding noise with variance (a) 0.1, (b) 1, (c) 5 and (d) 10 on the vanilla discriminator parameters. Data generated during training shows that the attractor is reasonably strong (a, b).

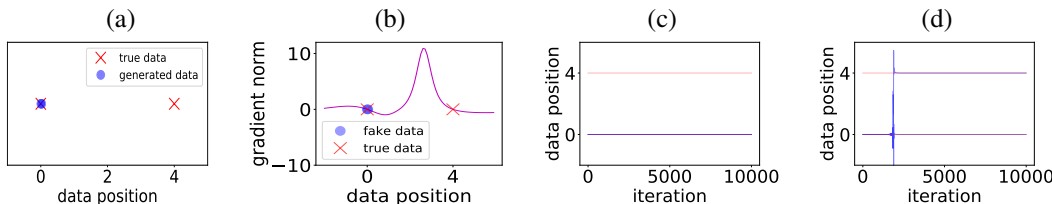

Figure 5: Escaping an attractor. We initialize the GAN at a local attractor (a) and illustrate the gradient norm in (b). The baselines get trapped at the initial point for the subsequent training (c), while the GAN with DigGAN regularizer can escape (d).

**Experiment 2: vanilla GAN cannot escape unregularized-local-attractors.** In this experiment, we verify that the end state $D_1$, $G_1$ is a stable local attractor for a vanilla GAN, by testing if adding noise to $D_1$ can aid GAN training escape and converge to a better result. We experiment with four levels of noise variance, 0.1, 1, 5 and 10 respectively. We visualize the data movement in Fig. 4.

We observe that 1) upon adding noise $\mathcal{N}(0, 0.1)$ the GAN's initial state differs from the local attractor, but GAN training eventually converges to the same attractor; 2) the GAN's initial state differs more from the local attractor after $\mathcal{N}(0, 1)$ noise is added, but GAN training eventually still converges to the same result; 3) Noise $\mathcal{N}(0, 5)$ has enough strength to help the GAN dynamics escape from the local attractor completely. However, GAN training is caught by another bad attractor, where both generated data points overlap at $x = 4$; 4) Noise $\mathcal{N}(0, 10)$ is too strong to make the dynamics converge. These experiments show that unregularized-local-attractors can affect vanilla GAN training.

**Experiment 3: DigGAN regularization helps to avoid unregularized-local-attractors.** To ensure that the comparison is fair, we train a DigGAN from the same starting point $D_0$ and $G_0$ as Experiment 1. We visualize the comparison with vanilla GAN in Fig. 3 bottom row. We observe that 1) DigGAN enables the GAN to avoid local attractors and cover two modes under the same settings (bottom row (a,b)). 2) Compared to the vanilla GAN whose gradient gap increases significantly before getting trapped, the proposed regularization $R$ reduces the risk of getting trapped by encouraging the gradient norm gap to remain at a low level (bottom row (c)). Note the scale difference of the y-axis.

**Experiment 4: DigGAN regularization helps to escape unregularized-local-attractors.** We demonstrate that the DigGAN regularization can help the dynamics escape unregularized-local-attractors. To observe this, we initialize the GAN at a state where it is trapped in a local attractor, which is denoted as $D_2$ and $G_2$. Starting from this initialization, a GAN without regularization remains trapped (see Fig. 4). In contrast, a GAN with DIG regularizer $R$ is able to escape eventually.

We provide more details regarding this experiment. At the initial state, the generated data covers one mode, and $\|\partial D/\partial x_R\|_2 - \|\partial D/\partial x_F\|_2| < 1e^{-5}$ (Fig. 5 bottom row). We initialize $G_2$ to be the final generator obtained when training the baseline in Experiment 2, which generates two points that overlap at $x = 0$. To get $D_2$, we fix $G_2$ and train the discriminator $D$ to optimality using the objective given in Eq. (3). This case is difficult because the proposed regularizer has little influence. However, because the discriminator's activation is non-linear, the discriminator aims to output a greater value for the missing point in subsequent training steps. This change in discriminator outputs causes a gradient gap, which disrupts the system's balance. Eventually, DigGAN is able to escape. We can detect a growing perturbation for the generated data around iteration 2000 (bottom row (d)).

|  | 100% data | | 20% data | | 10% data | |
|---|---|---|---|---|---|---|
|  | IS ↑ | FID ↓ | IS ↑ | FID ↓ | IS ↑ | FID ↓ |
| SNGAN | 8.42 | 19.29 | 7.24 | 30.69 | 6.39 | 47.08 |
| SNGAN + DigGAN (ours) | **8.54** | **15.11** | **7.78** | **22.27** | **7.36** | **29.43** |
| BigGAN | 9.04 | 10.53 | 8.26 | 21.38 | 7.62 | 36.35 |
| BigGAN + DigGAN (ours) | **9.09** | **9.74** | **8.40** | **17.11** | **7.89** | **23.75** |
| BigGAN + DiffAug | 9.04 | 9.88 | 8.66 | 14.51 | 8.19 | 23.65 |
| BigGAN + DiffAug + DigGAN (ours) | **9.28** | **8.49** | **8.89** | **13.01** | **8.32** | **17.87** |

Table 1: Inception score (IS) (higher is better) and Fréchet Inception distance (FID) (lower is better) for BigGAN trained with 100%, 20%, 10% CIFAR-10 data, respectively.

|  | 100% data | | 20% data | | 10% data | |
|---|---|---|---|---|---|---|
|  | IS ↑ | FID ↓ | IS ↑ | FID ↓ | IS ↑ | FID ↓ |
| BigGAN | 10.58 | 13.54 | 8.66 | 33.64 | 5.35 | 73.01 |
| BigGAN + $R_{LC}$ [51] | **11.18** | **11.88** | 9.08 | 25.51 | 7.76 | 49.63 |
| BigGAN + DigGAN (ours) | 10.92 | 12.93 | **9.21** | **21.79** | **9.06** | **27.61** |
| BigGAN + DiffAug | **11.87** | 12.00 | 9.41 | 22.14 | 8.63 | 33.70 |
| BigGAN + DiffAug + $R_{LC}$ [51] | 10.77 | 11.84 | 9.52 | 21.78 | 8.89 | 26.91 |
| BigGAN + DiffAug + DigGAN (ours) | 11.45 | **11.63** | **9.54** | **19.79** | **8.98** | **24.59** |

Table 2: Inception score (IS) (higher is better) and Fréchet Inception distance (FID) (lower is better) for BigGAN trained with 100%, 20%, 10% CIFAR-100 data, respectively.

# 4 Experiments

## 4.1 Experimental settings

To validate the effectiveness of the proposed DigGAN regularization, we conduct comprehensive experiments on CIFAR-10 [27], CIFAR-100, Tiny-ImageNet [30], CUB-200 [52] and multiple low-shot [61] data using both class-conditional BigGAN, the unconditional SNGAN and StyleGAN2. We use two standard evaluation metrics, i.e., the Fréchet Inception Distance (FID) [16] and the Inception Score (IS) [46]. A smaller FID and a larger IS generally imply better GAN models. Moreover, we further improve our empirical results by combining our regularization with DiffAug [61], a popular data-augmentation trick. For the regularization, we set $\lambda = 1000/$(data availability percentage) if no data augmentation is used; e.g., $\lambda = 5000$ if we train with 20% data. We set $\lambda = 10/$(data availability percentage)$^2$ if data augmentation is used. We also use $\alpha = 0.5$ in all real data experiments. For both conditional and unconditional GANs, we draw real samples and fake samples randomly in pairs (i.e., we pair samples irrespective of their class). All GANs are trained with 4 NVIDIA Geforce RTX 2080 Ti GPUs.

## 4.2 CIFAR-10 and CIFAR-100

We test our regularization on CIFAR-10 and CIFAR-100 data using two benchmark frameworks: BigGAN and SNGAN, following the implementation by Zhao et al. [61] and Miyato et al. [38]. Both CIFAR-10 and CIFAR-100 datasets have 50,000 training images with an image size of $32 \times 32$. CIFAR-100 is more challenging, since it includes 100 categories with less images per class, while CIFAR-10 has 10 categories. We conduct the experiments on 100%, 20%, 10% data, following the settings used in prior work [61, 51, 7]. We report the results in Tab. 1 and Tab. 2.

Four observations are notable. First, DigGAN can improve GAN performance generally, regardless of the amount of available data. Use of the regularization hence doesn't seem to harm GAN training. Second, our regularization can yield significant advantages when dealing with scarce data. Note that our regularization improves baseline FIDs by 11.85 score (33.64 vs. 21.79) and 45.4 score (73.01 vs. 27.61) for 20% and 10% CIFAR-100 data respectively. Third, with 10% CIFAR-100 data, the regularization can even outperform BigGAN that uses DiffAug by 6.09 FID (33.70 vs. 27.61). Fourth,

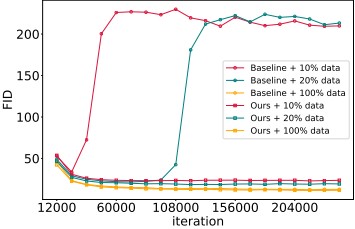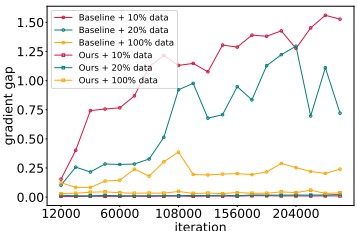

Figure 6: Comparison between BigGAN+DiffAug (baseline) and BigGAN+DiffAug+DigGAN regularization (ours) on CIFAR-100, trained with $100\%$, $20\%$ and $10\%$ training data. We show FID (left) and gradient norm gap (right) during training.

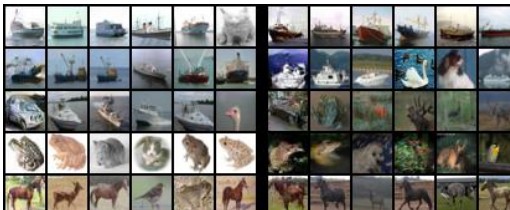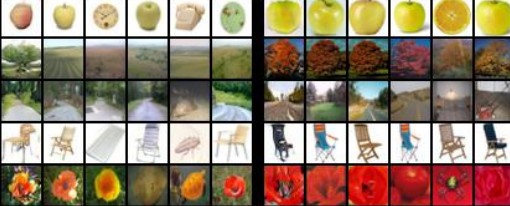

Figure 7: Generated images and their 5 nearest neighbors in the training data, with $10\%$ CIFAR-10 (left) and $10\%$ CIFAR-100 (right) training data.

DigGAN improves upon the regularization proposed by Tseng et al. [51] by 3.72 FID (25.51 vs. 21.79) and by 22.02 FID (49.63 vs. 27.61) in cases of $20\%$ and $10\%$ CIFAR-100 data respectively.

Data augmentation, a widely used trick for GAN training, is orthogonal to the studied regularization approach. To demonstrate orthogonality to data augmentation, we follow the experimental settings of DiffAug [61], a differentiable and data-level augmentation method. We report the results in Tab. 1 and Tab. 2 as well. Our technique can consistently enhance performance and improve upon baselines with the augmentation, regardless of data availability. The FID scores improve from 23.65 to 17.87 and from 33.70 to 24.59 for CIFAR-10 and CIFAR-100 with our regularization.

Besides Fig. 1, we also present the FIDs, gaps between the gradient of the discriminator's predictions on real images and generated images for CIFAR-100 with DiffAug in Fig. 6. We can draw similar conclusions: 1) For the baseline, the gaps get larger with increasingly scarcer data. 2) Our DigGAN regularization can control the discriminator gradient gaps at a low level. These observations align with our analysis in Sec. 3.

A possible concern relates to memorization of the scarce training data by the generator. To validate the generalization ability, we show the generated images and their 5 nearest neighbors from the training data, after training a generator with only $10\%$ training data (Fig. 7). We observe the generated data to not appear in the training data. This shows that the model doesn't just memorize.

### 4.3 Higher resolution generation with BigGAN: Tiny-ImageNet, CUB-200

Generating higher-resolution images when training with limited data is even more challenging. In this section, we report our results on Tiny-ImageNet and CUB-200. We adopt the implementation of the Pytorch StudioGAN codebase.[1] Tiny-ImageNet contains 100,000 images of 200 classes at a resolution of $64 \times 64$. We test our proposed approach using $10\%$, $50\%$, and $100\%$ of the data. CUB-200 data contains 5,994 training images of 200 classes at $128 \times 128$ resolution, and we use $50\%$ and $100\%$ of the data.

We compare with two baselines: BigGAN and $R_{LC}$ [51]. Tab. 3 lists all the quantitative results. We note that the conclusions are consistent with the CIFAR experiments. Specifically, 1) DigGAN outperforms the baseline when using all available data, regardless of the presence of the data augmentation; 2) DigGAN gains increase with fewer data. For instance, we improve $R_{LC}$ by 1.01

---

[1]https://github.com/POSTECH-CVLab/PyTorch-StudioGAN

|  | 100% Tiny ImageNet | 50% Tiny ImageNet | 10% Tiny ImageNet | 100% CUB-200 | 50% CUB-200 |
|---|---|---|---|---|---|
| BigGAN | 31.92 | 43.45 | 130.77 | 20.15 | 48.67 |
| BigGAN+$R_{LC}$ [51] | 28.11 | 36.11 | 121.16 | 40.37 | 98.38 |
| BigGAN + DigGAN (ours) | **17.76** | **24.63** | **84.27** | **14.45** | **23.20** |
| BigGAN + DiffAug | 16.33 | 24.50 | 95.40 | 13.49 | 24.35 |
| BigGAN+$R_{LC}$ [51]+DiffAug | 16.30 | 23.67 | 83.76 | 12.81 | 23.49 |
| BigGAN + DiffAug + DigGAN (ours) | **14.84** | **22.66** | **51.18** | **11.58** | **21.12** |

Table 3: Fréchet Inception distance (FID) for BigGAN with Tiny-ImageNet and CUB-200.

|  | 100-shot Obama | 100-shot grumpy cat | AnimalFace Dog | AnimalFace Cat |
|---|---|---|---|---|
| StyleGAN+ADA | 49.78 | 27.34 | 66.25 | 41.40 |
| StyleGAN+ADA+DigGAN | **41.34** | **26.75** | **59.00** | **37.61** |

Table 4: Fréchet Inception distance (FID) for StyleGAN2 with ADA on low-shot datasets.

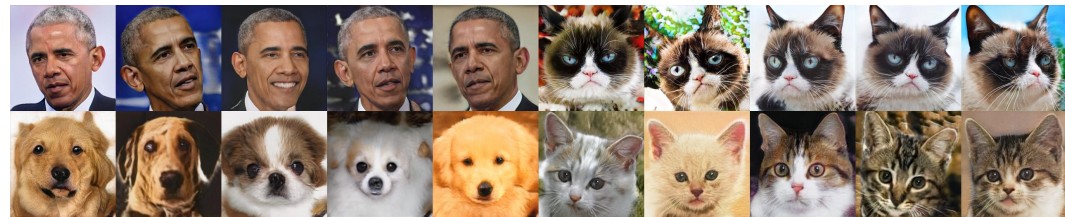

Figure 8: Low-shot generation with DigGAN on Obama, Grumpy Cat, and AnimalFace datasets.

FID (23.67 vs. 22.66) with 50% Tiny-ImageNet data, and increase the improvements to 32.58 FID (83.76 vs. 51.18) with 10% Tiny-ImageNet data.

## 4.4 Low-shot generation with StyleGAN2

We also test our DigGAN on low-shot generation tasks. We conduct new low-shot generation experiments with StyleGAN2+ADA [23]. We run experiments on the 100-shot Obama, 100-shot Grumpy Cat, and AnimalFace dataset (160 cats and 389 dogs) provided by [61]. All datasets are at $256 \times 256$ resolution. We use the maximum training length of 600k images for all experiments. We set the regularization weight as 100. We provide the results in Tab. 4 and show that DigGAN achieves consistent gains on all datasets. Generated images are in Fig. 8.

## 4.5 Comparison with other gradient-based regularization methods

Several gradient-based regularization methods have been proposed for GANs. Among them, GP-1 [15], R1, R2 [45] and DraGAN [37] are the most popular ones. However, GP-0 is usually used for an SNGAN structure [29], and R1 is usually applied in a StyleGAN2 framework [22]. R2 and DraGAN are not widely tested in popular model architectures. We provide a comparison on two settings: StyleGAN+CIFAR10 and BigGAN+CIFAR100. From Tab. 5, we observe: 1) GP-1 is not compatible with StyleGAN2, as it prevents StyleGAN2 from training properly. Further, it cannot be applied on BigGAN, since the classes are not available for the interpolated data. 2) DigGAN, R1, R2 and DraGAN can be applied to StyleGAN2 and BigGAN. With the full dataset available, DigGAN does not necessarily improve over other regularization methods. However, when limited training data are available, DigGAN is able to improve upon other gradient-based regularization techniques. Compared to the best baseline, DigGAN improves FID by 5.16 and 6.74 with 10% data availability.

## 4.6 Ablation study

**Regularization strength.** We conduct an ablation study regarding the regularization strength using 10% CIFAR-100 data and the BigGAN framework. We sweep the regularization strength $\lambda \in \{1, 10, 100, 1k, 10k, 100k\}$ and test the models' sensitivity. We report the results in Fig. 9. We observe that the performance improves with increasing strength at first, better and better addressing overfitting. However, as expected, the performance deteriorates when the regularization is too strong. To provide a more comprehensive comparison, we also sweep R1 regularization [45] over the

|  | StyleGAN2 + CIFAR-10 |  | BigGAN + CIFAR-100 |  |
|---|---|---|---|---|
|  | 100% | 10% | 100% | 10% |
| Baseline | – | – | 13.54 | 73.01 |
| +R1 | 11.07 | 36.02 | 15.63 | 40.62 |
| +R2 | **9.14** | 34.93 | 14.61 | 34.35 |
| +GP-1 | 200.51 | 228.39 | – | – |
| +DRAGAN | 9.56 | 39.21 | 13.09 | 52.79 |
| +DigGAN | 10.12 | **29.75** | **12.00** | **27.61** |

Table 5: Comparison with popular gradient-based regularization methods. We conduct experiments with two model architectures using two datasets, and report FID.

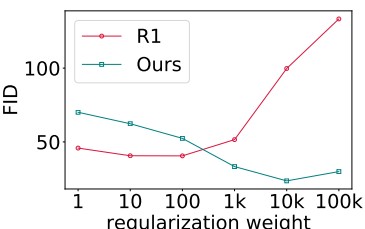

Figure 9: An ablation study of our regularization and R1 [45] over regularization strength.

|  | 100% CIAFR-10 | 10% CIAFR-10 | 100% CIAFR-100 | 10% CIAFR-100 |
|---|---|---|---|---|
| Random pairing (Ours) | 9.74 | 23.75 | 12.93 | 27.61 |
| Same-class pairing | 10.12 | 21.53 | 13.14 | 28.08 |
| Gradient magnitude pairing | 9.09 | 27.64 | 12.86 | 47.75 |

Table 6: Ablation study on real and fake samples pairing mechanism. All experiments are conducted with BigGAN structure.

strength set and report the results in Fig. 9. The behavior of R1 is similar to the proposed DigGAN regularization, however, its best quantitative results tend to be lower.

**Pairing mechanism.** We draw real samples and fake samples randomly in pairs and irrespective of their class for our experiments. To study the effect of the pairing method on the performance, we compare random pairing with two other methods: pairing within the same class and pairing based on gradient magnitude order. In this ablation study, we conduct the comparison using CIFAR data and BigGAN structure. We use the default regularization weight. We report results in Tab. 6.

For the same-class pairing mechanism, each pair of the real sample and fake sample is drawn from the same class. Results are shown in Tab. 6 row 2: we do not observe a significant difference to random pairing. We think that the use of exponential moving average (EMA) ensures that the trained model is not significantly influenced by one pair. For the gradient magnitude pairing mechanism, we pair real samples and fake samples by sorting norms $\|\partial D/\partial x_R\|_2$ and $\|\partial D/\partial x_F\|_2$ within each batch to show how permutations of the real and fake samples affect the results. Results are in Tab. 6 row 3. We observe that gradient magnitude pairing works better with 100% data availability for both datasets. However, DigGAN works much better than gradient magnitude pairing with 10% data availability. We think random pairing is important.

## 5  Conclusion and Broader Impact

We introduce DigGAN, a gradient-based regularization for the discriminator which improves GAN training with limited data. DigGAN encourages a small difference between the gradient norm of the discriminator w.r.t. the real data and w.r.t. the generated data. Empirically, we observe this difference to be large when GAN training fails. We also demonstrate that DigGAN regularization reduces the effect of local attractors. On a variety of datasets and architectures we show consistent results.

**Limitation.** We tried to analyze DigGAN from a theoretical perspective. But a theoretical analysis of the studied loss is extremely challenging because of the way batching is performed. We could not find a compelling theoretical result that is meaningful, and we did not want to include derivations that don't add insights.

**Societal implications.** This work can have both positive and negative effects. On the positive side, training GANs with limited real-world data is important for fields where it is expensive to collect large datasets. For instance, our research can make AI methods more powerful in rare disease diagnosis, antique authentication, etc. On the negative side, improved generative methods can be used to generate fake data and spread misinformation via DeepFakes.

**Acknowledgements.** Work supported in part by NSF under Grants 1718221, 2008387, 2045586, 2106825, MRI 1725729, and NIFA award 2020-67021-32799. We thank NVIDIA for a GPU.

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
