# OpenReview forum: "DigGAN: Discriminator gradIent Gap Regularization for GAN Training with Limited Data"
_NeurIPS.cc/2022/Conference — NeurIPS 2022 Accept_

### Official Review · Reviewer_Z5nH · 2022-07-08

**Rating:** 6
**Confidence:** 5
**Soundness:** 2 fair
**Presentation:** 3 good
**Contribution:** 3 good

**Summary:**

This paper introduces a new data-efficient training technique for GAN. The technique named Discriminator gradIent Gap (DIG) regularization aims to equalize the changes of discriminator’s judgments w.r.t its inputs (real and fake data) so that the discriminator can effectively learn representations for adversarial training without an imbalance issue. Experimental results demonstrate that GAN with the proposed DIG regularization exhibits better image generation results than the previous techniques (LeCam regularization and DiffAug).

**Questions:**

[Q1] Which gradient (gradient of a discriminator's prediction w.r.t real or fake sample) makes the DIG value large? I guess the norm of the gradient w.r.t real sample might be very large, as the discriminator tends to memorize training samples and predicts samples other than the training samples to be fake regardless of the realism of samples.

**Ethics Review Area:**

["I don’t know"]

**Limitations:**

I cannot find any section that explains the limitations of the paper. Also, the described negative societal impact is not sufficient. I hope the authors will add more portions to tell the pros and cons of the paper and its societal impact.

**Strengths And Weaknesses:**

Strengths

[S1] I think the proposed method is new to the GAN community.

[S2] The empirical finding that the gap between the norms of the gradients of discriminator’s predictions increases when the GAN is trained with fewer data, seems to be very useful for future development of data-efficient training methods.

[S3] The experiments in the paper show that the proposed DIG regularization is helpful for GAN training in data-hungry situations. To demonstrate the effectiveness of DIG regularization, the authors utilize various datasets (CIFAR10, CIFAR100, Tiny ImageNet, and CUB200), which means the proposed method can be used for real-world applications.

[S4] The paper is well written and easy to follow.

Weaknesses

[W1] I think the theoretical explanation of the relation between DIG and the training failure is needed. Although experimental results show that applying DIG regularization can resolve the problem of getting stuck in local attractors, I am not sure why this is possible by imposing DIG regularization.

[W2] FID scores in this paper do not match the results of previous papers [1, 2]. The FIDs of BigGAN on CIFAR10 in the papers [1, 2] are 8.57 and 8.08, respectively, while the FID of BigGAN in this paper is 10.53 (Table 1). Performance discrepancies occur for experiments across the paper, which significantly lowers the credibility of the paper.

[W3] Experiments using high-resolution images are missing. Verifying the usefulness of DIG regularization using MetFaces, AFHQv2, or FFHQ will make the paper more convincing.

[W4] Comparision with Adaptive Discriminator Augmentation (ADA) is missing. Since ADA is a standard method to train GAN with limited data, I think comparing DIG regularization with ADA is necessary.

[1] Chen, T., Cheng, Y., Gan, Z., Liu, J., & Wang, Z. (2021). Ultra-Data-Efficient GAN Training: Drawing A Lottery Ticket First, Then Training It Toughly. ArXiv, abs/2103.00397.
https://openreview.net/forum?id=BBVcs78PEDA

[2] Kang, M., Shim, W., Cho, M., & Park, J. (2021). Rebooting ACGAN: Auxiliary Classifier GANs with Stable Training. NeurIPS.

---

> ### Author Response · Authors · 2022-08-02
> **Response**
>
> **Question: Theoretical explanation of the relation between DIG and the training failure.**
>
> **Answer:** We tried to analyze DigGAN from a theoretical perspective for quite a while actually. But a theoretical analysis of the studied loss is extremely challenging because of the way batching is performed. We could not find a compelling theoretical result that is meaningful, and we didn’t want to include derivations that don’t add insights.
>
> &nbsp;
> &nbsp;
>
> **Question: FID scores in this paper do not match the results of previous papers [1, 2].**
>
> **Answer:** Thanks for pointing this out. We use a different version of the code from [1, 2]. [1, 2] use the StudioGAN codebase for DiffAug, while we use the original codebase provided by the DiffAug authors (https://github.com/mit-han-lab/data-efficient-gans). Thus, compared to [1,2], our results are closer to [a].  Concretely, FIDs are 9.59, 21.58, and 39.78 for 100%, 20% and 10% data availability in [a], and ours are 10.53, 21.38 and 36.35 for 100%, 20% and 10% data availability. The difference in the case of 100% data availability is due to the randomness in training and evaluation. More importantly, our baseline results for 20% and 10% data availability are better than the ones reported in the baseline [a]. Since limited data is the main focus, we opted for the stronger baseline numbers.
>
> [a] S. Zhao, Z. Liu, J. Lin, J.-Y. Zhu, and S. Han. Differentiable augmentation for data-efficient gan training. In Advances in Neural Information Processing Systems, 2020.
>
> &nbsp;
> &nbsp;
>
> **Question: Experiments using high-resolution images are missing.**
>
> **Answer:** We provide 256x256 resolution image generation results. Larger-scale experiments are beyond our compute budget.
> Specifically, we conduct new low-shot generation experiments with StyleGAN-V2+ADA. We run experiments on the 100-shot Obama, 100-shot Grumpy Cat, and AnimalFace dataset (160 cats and 389 dogs) provided by [a]. All datasets have a 256×256 resolution. We use the maximum training length of 600k images for all experiments. We set the regularization weight=100. We provide the results below. Results show that DigGAN achieves consistent gains on all datasets.
>
> |                     | 100-shot Obama | 100-shot grumpy cat | AnimalFace Dog  | AnimalFace Cat |
> |---------------------|----------------|---------------------|-----------------|----------------|
> | StyleGAN+ADA        | 49.78          | 27.34               | 66.25           | 41.40          |
> | StyleGAN+ADA+DigGAN | **41.34**          | **26.75**               | **59.00**           | **37.61**          |
>
> &nbsp;
> &nbsp;
>
> **Question: Comparision with Adaptive Discriminator Augmentation (ADA) is missing.**
>
> **Answer:** Please see the table above for comparison to ADA.
>
> &nbsp;
> &nbsp;
>
> **Question: Which gradient (gradient of a discriminator's prediction w.r.t real or fake sample) makes the DIG value large?**
>
> **Answer:** We don’t observe any specific gradient norm magnitude pattern, regardless of dataset and model structure. For instance,
> - The gradient norm for fake data is larger for SN-GAN+10%CIFAR10.
> - The gradient norm for real data is larger for BigGAN+20%CIFAR10/CIFAR100.
> - The gradient norm for real data is larger for BigGAN+10%CIFAR10/CIFAR100.
> - The gradient norm for fake data is larger for BigGAN+50% TinyImageNet.
> - The gradient norm for real data is larger for BigGAN+10% TinyImageNet.
>
> &nbsp;
> &nbsp;
>
> **Question: Missing discussion of limitations, and insufficient discussion of societal impact.**
>
> **Answer:** One limitation of our work: results still need to be improved for very scarce data. (e.g. 10% Tiny ImageNet, FID=84.27 without DiffAug and 51.18 with DiffAug). We hope our work encourages more research in this direction.
>
> We update the social impact as follows. This work can have both positive and negative effects. On the positive side, training GANs with limited real-world data is important for fields where it is expensive to collect large datasets. For instance, our research can make AI methods more powerful in rare disease diagnosis, antique authentication, etc. On the negative side, improved generative methods can be used to generate fake data and spread misinformation via DeepFakes.

---

> > ### Comment · Reviewer_Z5nH · 2022-08-07
> > **Thank you. I'll raise the score from 4 to 6.**
> >
> > **Question: Theoretical explanation of the relation between DIG and the training failure.**
> >
> > => Thanks for the kind explanation, but this does not resolve my concern (W1).
> >
> > **Question: FID scores in this paper do not match the results of previous papers [1, 2].**
> >
> > => Thank you! This resolve my concern (W2).
> >
> > **Suggestion**: Refer to Figure 6 and Table 7 in the paper (https://arxiv.org/abs/2206.09479?context=cs) to compare DigGAN with DiffAugment, ADA, LeCam, and APA.
> >
> > **Question: Experiments using high-resolution images are missing, Comparision with Adaptive Discriminator Augmentation (ADA) is missing.**
> >
> > => I compared DigGAN's results with values in DiffAugment paper (page 8). DigGAN seems useful when trying to train a GAN in situations where data is scarce.
> >
> > **Question: Which gradient (gradient of a discriminator's prediction w.r.t real or fake sample) makes the DIG value large?**
> >
> > => I believe providing the source of a large DIG value makes the authors' paper more convincing.
> >
> > **Question: Missing discussion of limitations, and insufficient discussion of societal impact.**
> >
> > => Good. Thank you.
> >
> > Overall, I am satisfied with the explanations provided by the authors. Some of the concerns I raised are not fully addressed, but I believe this paper has enough quality to be accepted at Neurips. I will raise the score from 4 to 6.
> >
> > Cheer,
> >
> > Reviewer Z5nH

---

### Official Review · Reviewer_Mjjy · 2022-07-11

**Rating:** 5
**Confidence:** 5
**Soundness:** 2 fair
**Presentation:** 3 good
**Contribution:** 2 fair

**Summary:**

This paper proposes a new method for improving gan training under limited data. The key idea is to regularize the gradient norm of discriminator between real and fake samples. Authors have done empirical studies showing that this regularization can help avoid bad attractors within the GAN loss landscape. Authors have applied the proposed regularization on BigGAN, leading to improvement across multiple datasets.

**Questions:**

Overall I think the idea proposed in this manuscript looks interesting and effective to some extent. But I think the experiments are insufficient to validate the effectiveness of the proposed method. Detailed questions are covered in paper weaknesses.

**Limitations:**

Authors didn't include the discussion of limitations in the main manuscript. I think authors can briefly discuss in what cases the proposed method may fail or lead to marginal improvement.


**Strengths And Weaknesses:**

Strengths:
+ The manuscript is easy to follow
+ The empirical studies provided by the authors are important to understand the effectiveness of proposed regularization

Weaknesses:
- Missing related work, and incorrect interpretation of related work. Specifically, [b] ([17] in manuscript) is not doing data augmentation in my understanding. And [a] is also related but is missed in the manuscript. Since both [a] and [b] are not doing data augmentation, I believe they should be compared as well.

[a] Data-Efficient Instance Generation from Instance Discrimination
[b] Deceive d: Adaptive pseudo augmentation for gan training with limited data.

- while at L30 authors claim "data augmentation can enhance the results, but the benefit is limited with insufficient data (Tab. 3)." In table 3 authors didn't evaluate sufficient data augmentation methods to support this claim.

- in the experiments, authors only compared to DiffAug and R_LC, which is insufficient. Authors should also include ADA, etc.

- while previous works targeting gan training with limited data mainly evaluate their methods on StyleGAN and unconditional image synthesis, authors mainly evaluate the proposed method on BigGAN and conditional image synthesis. It's better to stay consistent with previous methods to ensure a fair comparison. Or at least expand table 4.

-  It's great that authors provide empirical studies in Sec.3.3. It's better if authors include a theoretical analysis.

---

> ### Author Response · Authors · 2022-08-02
> **Response**
>
> **Question: Missing and incorrect interpretation of related work. [a] Data-Efficient Instance Generation from Instance Discrimination; [b] Deceive d: Adaptive pseudo augmentation for gan training with limited data.**
>
> **Answer:** Thanks a lot for pointing out [a]. We will discuss [a] in the related work section. Our approach differs from [a]: we perform binary classification (i.e., real vs. fake) and propose a new regularizer that encourages the Discriminator gradIent Gap of a GAN to be small. In contrast, [a] requires the discriminator to do instance-level classification (i.e. distinguish every individual real and fake image instance as an independent category).
> For [b], we understand it uses generated data as an “augmentation” for the real data (see the paper title which mentions the word “augmentation”). Hence, we think it can be referred to as performing “augmentation”. We will clarify this in the related work section.
>
> &nbsp;
> &nbsp;
>
> **Question: Insufficient experiments on ADA and StyleGAN-v2, which are the main focus of prior work.**
>
> **Answer:** We don’t think prior work mainly focuses on StyleGAN and unconditional image synthesis. E.g., [1, 2, 3] conduct the majority of their experiments with BigGAN and conditional image synthesis.
> To provide additional evidence, we conduct new low-shot generation experiments with StyleGAN-V2+ADA. We run experiments on the 100-shot Obama, 100-shot Grumpy Cat, and AnimalFace dataset (160 cats and 389 dogs) provided by [3]. All datasets are at 256×256 resolution. We use the maximum training length of 600k images for all experiments. We set the regularization weight=100. We provide the results below. Results show that DigGAN achieves consistent gains on all datasets.
>
> |                     | 100-shot Obama | 100-shot grumpy cat | AnimalFace Dog  | AnimalFace Cat |
> |---------------------|----------------|---------------------|-----------------|----------------|
> | StyleGAN+ADA        | 49.78          | 27.34               | 66.25           | 41.40          |
> | StyleGAN+ADA+DigGAN | **41.34**          | **26.75**               | **59.00**           | **37.61**          |
>
> [1] H.-Y. Tseng, L. Jiang, C. Liu, M.-H. Yang, and W. Yang. Regularizing generative adversarial networks under limited data. In Proceedings of CVPR, 2021.
>
> [2] T. Chen, Y. Cheng, Z. Gan, J. Liu, and Z. Wang. Data-efficient gan training beyond (just) augmentations: A lottery ticket perspective. In Neurips, 2021.
>
> [3] S. Zhao, Z. Liu, J. Lin, J.-Y. Zhu, and S. Han. Differentiable augmentation for data-efficient gan training. In Neurips, 2020.
>
> &nbsp;
> &nbsp;
>
> **Question: Theoretical analysis in Sec. 3.3.**
>
> **Answer:** We tried to analyze DigGAN from a theoretical perspective for quite a while actually. But a theoretical analysis of the studied loss is extremely challenging because of the way batching is performed. We could not find a compelling theoretical result that is meaningful, and we didn’t want to include derivations that don’t add insights.
>
> &nbsp;
> &nbsp;
>
> **Question: Insufficient experiments to validate the effectiveness of the proposed method.**
>
> **Answer:** We kindly disagree that experiments are insufficient. We provide a significant amount of experiments on both synthetic and real data to show the efficacy and efficiency of DigGAN.
>
> &nbsp;
> &nbsp;
>
> **Question: Missing discussion of limitations.**
>
> **Answer:** One limitation of our work: results still need to be improved for very scarce data. (e.g. 10% Tiny ImageNet, FID=84.27 without DiffAug and 51.18 with DiffAug). We hope our work encourages more research in this direction.

---

> > ### Comment · Reviewer_Mjjy · 2022-08-08
> > **Response**
> >
> > I appreciate authors' effort on addressing my concerns. After checking the responses, as well as other reviewers' comments and authors feedback, my concerns have been well addressed.

---

### Official Review · Reviewer_ipzv · 2022-07-13

**Rating:** 6
**Confidence:** 3
**Soundness:** 3 good
**Presentation:** 3 good
**Contribution:** 3 good

**Summary:**

The authors propose a novel approach to training GANs with limited data, one that is motivated by the gap in the norms of the gradients between the target and generated images. They demonstrate applications of the proposed regularizer on various BigGAN architectures with and without data augmentation.

**Questions:**

 - **Loss Formulation:** I have a fundamental concern with the form of the loss $\left(  \left\| \frac{\partial D}{\partial x_R} \right\|  - \left\| \frac{\partial D}{\partial x_F} \right\|\right)$. It is unclear how the samples are drawn. If $x_R$ and $x_F$ are drawn randomly in pair, then, the particular paining of the real and fake images will affect how this loss performs. While this issue might not be visible in the synthetic 1D experiment present, on the image scale, this might have effect. For example, are we making sure the pairs are drawn from the same class. In class-conditional GANs, it might be possible, bit, in unconditional variants, this cannot be guaranteed, and there is a strong possibility that the manifold nature of each class might itself affect how these terms interact.
 - To address the above, maybe the authors could consider some ablation study where, given the choice of the sample batch, how permutations of the real and fake samples affects the penalty?
 - Alternatively, have the authors considered batch-level sample averages of the gradient norms? This would essentially bring the terms closed to the R1 and R2 penalties.

**Limitations:**

—

**Strengths And Weaknesses:**

**Originality and Significance:** Limited data (LD) training with GANs has been gaining popularity in the recent years, and the proposed regularization approach to training shows potential. The fact that it can be used in parallel with other schemes for LD training such as DiffAug is an advantage. The gradient gap in the discriminator is generally well analyzed and adequate  experimental validations are provided.

**Presentation and Clarity:** The paper is clear to read and the flow is consistent.
 - The synthetic experiments with Vanilla GAN consider a simplistic scenario — a similar experiment in 2D might have been more insightful, as the effect of different norm on the gradient vector (L1, L2), is clearer at least in 2D. Nevertheless, the given experiments help to give a clear intuition on the proposed method.
 - The notation is confusing at times. For example, The additional parentheses when time indexing time $t$ seems unnecessary, but this is just a personal opinion.
 - Some proof read would help, as there are a few typos here and there. L233 L146 Forth -> Fourth. L262 improvement to 32.56 -> improvement **by** 32.58, just to name a few.

**Literature Survey:** Most of the relevant literature has been cited and the paper includes relevant discussions on related works.

---

> ### Author Response · Authors · 2022-08-02
> **Response**
>
> **Question: How do permutations of the real and fake samples affect the penalty?**
>
> **Answer:** In our answer to the previous question, we studied class-conditional pairing and didn’t find a significant difference.
> In addition, here, we show how permutations of the real and fake samples affect the results. Specifically, we pair ||∂D/∂xR|| and ||∂D/∂xF|| by sorting the norms within each batch instead of pairing randomly. We conduct the comparison experiments on CIFAR-10 and CIFAR-100 with the BigGAN structure. We use the default regularization weight. We observe that the “sorting norm permutation” works better with 100% data availability for both datasets. However, DigGAN works much better than the “sorting norm permutation” method with 10% data availability. We think the random pairing is important.
>
> |                                   | 100% CIFAR-10 | 10% CIFAR-10 | 100% CIFAR-100 | 10% CIFAR-100 |
> |-----------------------------------|---------------|--------------|----------------|---------------|
> | random pairing (DigGAN)           | 9.74          | **23.75**        | 12.93          | **27.61**         |
> | sorting norm permutation  pairing | **9.09**          | 27.64        | **12.86**          | 47.75         |
>
> &nbsp;
> &nbsp;
>
> **Question: Have the authors considered batch-level sample averages of the gradient norms?**
>
> **Answer:** Great question. Indeed, before writing the paper we considered many gradient penalty variants. But none worked as well as the one reported in the paper. We provide regularization variants and results below. For each variant, we search the weight \lambda from the set {0.1, 1, 10, 100, 10^3, 10^4, 10^5}.
>
> | Regularization format                                                                 | 100% CIFAR-100 FID | 10% CIFAR-100 FID |
> |---------------------------------------------------------------------------------------|--------------------|-------------------|
> | DigGAN: ( (\|\|∂D/∂xR\|\|_2 - \|\|∂D/∂xF\|\|_2) **2 ).mean() * \lambda                | 12.93              | **28.61**             |
> | Variant 1: ( (\|\|∂D/∂xR - ∂D/∂xF\|\|_2) **2 ).mean() * \lambda                       | **12.74**              | 32.97             |
> | Variant 2: ( (\|\|∂D/∂xR\|\|_2^2).mean() - (\|∂D/∂xF\|\|_2^2).mean()) **2 ) * \lambda | 32.97              | 32.97             |
> | Variant 3: ( (\|\|∂D/∂xR\|\|_2).mean() - (\|\|∂D/∂xF\|\|_2).mean()) **2 ) * \lambda   | 12.40              | 50.75             |

---

> > ### Comment · Reviewer_ipzv · 2022-08-08
> > **Re: Response**
> >
> > Thank you for the detailed responses to my questions, and considering including the additional 2D experiments as parts of the supplementary. The main concern I had regarding the approaches to pairing data and how it affects the performance appear to be answered. It seems that there is some variability in performance based on how these pairs and picked and evaluated, depending also, on the kind of training conditions (say, data availability, etc).
> >
> > I think the authors could include a summary of these results as part of the experimental results and ablation study in Sec. 4, and consider including the detailed analysis as part of the Supplementary, as some of these questions might come up when others read the paper as well. The results and observations of the paper overall are novel and insightful, with applications to multiple GAN flavors, and I am still in favor of accepting the paper. I am raising my score from a 5 to a 6.

---

> ### Author Response · Authors · 2022-08-02
> **Response**
>
> **Question: 2-D synthetic experiments**
>
> **Answer:** We use 1-D synthetic experiments to visualize as many details as possible. To answer the question, we conducted four new 2-D synthetic experiments. The results are provided below and they are consistent with the results of the 1-D experiments in the paper.
> In all four experiments, we consider the task of generating a 5-Gaussian mixture with modes evenly spaced on a circle of radius 1. The generator is a 4-layer fully connected network (FCN) with 8 neurons per layer and leaky ReLU activation. The discriminator is a 4-layer FCN with 128 neurons per layer and ReLU activation. The experiments follow the same logic as the 1-D experiments in the paper. All experiments are run for 30k iterations.
>
> For each result, we provide 3 plots: 1. real data and generated data distribution, 2. logits: sigmoid(D(x)) distribution in the 2D space, 3. dD/dx distribution in the 2D space.
>
> Experiment 1: vanilla GAN can get stuck in unregularized-local-attractors. We train a vanilla GAN from scratch with a random initialization D0 and G0. We get D1 and G1 at the end of training. D1 and G1 end up covering only two of the five clusters (bad local attractor). [exp1 repository: https://anonymous.4open.science/r/DigGAN_rebuttal-A8B7/exp1]
>
> Experiment 2: vanilla GAN cannot escape unregularized-local-attractors. We verify that the state D1, G1 is a stable local attractor for a vanilla GAN, by testing if adding noise to the generated data from the beginning would lead to a successful escape. We experiment with three levels of noise variance: 0.1, 1, and 10. None of them help the vanilla GAN escape from the bad local attractor. [exp2 repository: https://anonymous.4open.science/r/DigGAN_rebuttal-A8B7/exp2]
>
> Experiment 3: DigGAN regularization helps to avoid unregularized-local-attractors. We train a DigGAN from the same starting point D0 and G0 used in Experiment 1. We observe that DigGAN ends up covering all 5 clusters at the end of training [exp3.1 repository: https://anonymous.4open.science/r/DigGAN_rebuttal-A8B7/exp3.1]. We also train DigGAN starting from D1 and G1 without adding noise. DigGAN also escapes from the bad local attractor and converges to a good global attractor in the end [exp3.2 repository: https://anonymous.4open.science/r/DigGAN_rebuttal-A8B7/exp3.2].
>
> Experiment 4: DigGAN regularization helps to escape unregularized-local-attractors. Following the setting in our paper, we initialize with D2 and G2 which only covers 2 out of 5 clusters and for which  | ||∂D/∂x||_2 −||∂D/∂xF||_2| < 1e^−2. Training a DigGAN from D2 and G2, we observe that DigGAN ends up escaping and covering all 5 clusters [exp4 repository: https://anonymous.4open.science/r/DigGAN_rebuttal-A8B7/exp4].
>
> To sum up, the observations in this 2-D experiment are consistent with the observations in the 1-D experiment. We think 1-D experiments provide cleaner visualizations. We will add these 2-D experiments to the appendix.
>
> &nbsp;
> &nbsp;
>
> **The notation is confusing at times, but this is just a personal opinion**
>
> Thanks a lot for the suggestion, we’ll simplify.
>
> &nbsp;
> &nbsp;
>
> **Some proofreading would help.**
>
> Thanks a lot, we’ll fix it.
>
> &nbsp;
> &nbsp;
>
> **Question: How are real and fake samples drawn in each batch, randomly or from the same class? How does pairing of the real and fake data affect results?**
>
> **Answer:** Great observation. We draw real samples and fake samples randomly in pairs, and we studied the pairing effect at length before submission too: 1) To reduce the effect of a particular pair on the loss, we use the exponential moving average (EMA) (line 127), which ensures that the trained model isn’t significantly influenced by one pair. We observe EMA helps to significantly stabilize the training. 2) Other losses which we considered to mitigate this effect, e.g., difference of sample-averaged norms, didn’t yield results that were as good as random pairing.
>
> Note, for all experiments, we pair samples irrespective of their class, for both conditional GAN and unconditional GAN. We also run experiments where we draw real and fake samples from the same class for CIFAR-100. We provide the results below, which exhibit no significant difference. This is perhaps due to the use of EMA.
>
> |                         | 100% CIFAR-10 | 10% CIFAR-10 | 100% CIFAR-100 | 10% CIFAR-100 |
> |-------------------------|---------------|--------------|----------------|---------------|
> | random pairing (DigGAN) |  **9.74**         | 23.75        | **12.93**          | **27.61**         |
> | same-class pairing      | 10.12         | **21.53**        | 13.14          | 28.08         |
>
> &nbsp;
> &nbsp;

---

### Meta-Review · Area_Chair_1r2D · 2022-08-30

**Recommendation:** Accept
**Confidence:** Less certain

**Metareview:**

The paper proposes a regularizer for limited-data GAN training. All three reviewers thought the experiments were adequate to demonstrate the method's usefulness and the writing was clear. The paper's biggest weakness seems to be unconvincing conceptual intutition and lack of theoretical justification (pointed out by reviewers Mjjy and Z5nH). This is a borderline paper but I recommend acceptance.

**Award:**

No

---

### Decision · Program_Chairs · 2022-09-14

Accept